# Enhancing Guided Bone Regeneration with a Novel Carp Collagen Scaffold: Principles and Applications

**DOI:** 10.3390/jfb15060150

**Published:** 2024-06-01

**Authors:** Michele Bujda, Karel Klíma

**Affiliations:** Department of Oral and Maxillofacial Surgery, 1st Faculty of Medicine and General University Hospital in Prague, Charles University, 12108 Prague, Czech Republic

**Keywords:** bone regeneration, tissue engineering, GBR membrane, carp collagen, bioactive scaffold

## Abstract

Bone defects resulting from trauma, surgery, and congenital, infectious, or oncological diseases are a functional and aesthetic burden for patients. Bone regeneration is a demanding procedure, involving a spectrum of molecular processes and requiring the use of various scaffolds and substances, often yielding an unsatisfactory result. Recently, the new collagen sponge and its structural derivatives manufactured from European carp (*Cyprinus carpio*) were introduced and patented. Due to its fish origin, the novel scaffold poses no risk of allergic reactions or transfer of zoonoses and additionally shows superior biocompatibility, mechanical stability, adjustable degradation rate, and porosity. In this review, we focus on the basic principles of bone regeneration and describe the characteristics of an “ideal” bone scaffold focusing on guided bone regeneration. Moreover, we suggest several possible applications of this novel material in bone regeneration processes, thus opening new horizons for further research.

## 1. Introduction

Bone regeneration procedures are part of tissue engineering and regenerative medicine (TERM), the main goal of which is the efficient replacement and/or regeneration of damaged tissue with an appropriate scaffold [1].

In clinical practice, new bone formation is achieved by various augmentation techniques with the use of different biomaterials to substitute bone itself and provide an optimal niche for healing.

There are a number of products available on the market with the aim of providing satisfactory results with respect to the bone volume and quality gained together with favorable handling properties and price. Moreover, the risk of immunogenic and allergic reactions or even the transmission of infectious diseases cannot be overlooked. Such demands are very difficult to meet in one product, as autografts, allografts, xenografts, and alloplastic materials have their own particular advantages and disadvantages. Therefore, there is usually a need for product combinations in clinical practice, and yet there is still no single product or fixed combination that serves as the gold standard for all types of bone defects.

Fish-derived tissue scaffolds seem promising due to their biocompatibility [2] and the ability to promote cell adhesion, as well as their favorable immunomodulatory properties [3]. They pose practically no threat of transmissible disease or religious objections. Additionally, they are extremely cost-effective, as they can be manufactured from waste (skin, scales, heads, fins, bones, etc.) [4].

The new collagen material derived from the skin of freshwater fish *Cyprinus carpio* has several structural variants, all registered under Patent No. CZ308862B6 [5]. This material has already been tested in vitro and in vivo with promising results and has also been suggested for use as a wound dressing, surgical site covering, vascular graft coating, and drug delivery system.

In this review, we focus on the main biological principles of bone regeneration, the necessities required for a successful procedure, and the ideal material characteristics used in guided bone regeneration (GBR). We introduce possible uses of novel collagen material in bone augmentation techniques and advocate its potential advantages over currently commercially dominant mammalian xenograft materials. However, this new carp-derived collagen scaffold has its limitations (discussed in this review) and was primarily designed as a wound coverage and drug delivery system. Therefore, we also hypothesize an eventual “improvement” in the novel material to refine its potential in relation to bone regeneration specifically.

As noted by Maganathan et al. [6], despite ongoing research, there is currently no commercial fish-derived scaffold available for human use in regenerative medicine. Therefore, there is an urgent need to fill this market gap and take advantage of fish products, known for their high healing rate.

## 2. Methods

A comprehensive literature search was conducted using PubMed, Web of Science, and the Cochrane Library to identify relevant studies. During the search, we used the following keywords: “*bone regeneration*”, “*guided bone regeneration*”, “*GBR*”, “*fish collagen*”, “*freshwater fish collagen*”, “*carp collagen*”, “*GBR membrane*”, “*collagen scaffold*”, and “*bone augmentation*”.

In total, 129 references were selected for inclusion in this review. These include journal articles and books focusing on bone regeneration procedures and implant dentistry, and finally Patent No. CZ308862B6. From this document, we acquired information about novel carp-derived collagen material and its derivates. The selection of references was based on their relevance to the topic of bone regeneration with a focus on guided bone regeneration and the topic of fish-derived collagen scaffolds and their use in TERM.

## 3. Bone Regeneration

Bone tissue engineering is a promising technique developed to regenerate lost bone due to trauma, tumor, infection, connective tissue disease, congenital defects, surgery [7] or even physiological atrophy to rehabilitate patients in a functional and aesthetic manner.

Knowledge of bone structure, bone healing physiology, awareness of surgical techniques, and available biomaterials for bone regeneration purposes is necessary for predictable and successful bone gain in the desired area.

### 3.1. Bone Structure

Bone, the mineralized connective tissue, comprises organic and inorganic components, of which collagen (25–30% of all proteins in the human body) and hydroxyapatite are the most abundant [8].

Bone, except for its mechanical and supportive function, is a deposit of minerals, a source of progenitor stem cells, and an organ responsible for hematopoiesis.

The strength of bone is directly related to its mineral content [9] but also depends on its intrinsic organization. According to the internal structure and mechanical characteristics, bone tissue can be classified into *woven bone* and *lamellar bone.* Misch et al. [10] described an intermediate type of bone, i.e., *composite bone*, which was initially observed and described by Enlow et al. [11,12] (Appendix A). Lamellar bone is further classified as *cancellous* (also trabecular or spongy) bone or *cortical* (or compact) bone according to lamellae organization.

### 3.2. Bone Healing and Remodeling

When bone sustains any injury or trauma, it undergoes a physiological process to restore its function and continuity, referred to as **bone healing**, which can take two distinct forms depending on the specific conditions present.

*Primary healing,* also known as direct healing, occurs in situations where individual bone fragments are perfectly aligned (without any gap) and there is practically no movement between fragments. These conditions provide absolute stability of the fracture site, allowing healing through direct **bone remodeling** (physiological bone turnover) through so-called **cutting cones** resulting in restoration of osteons [13] (Figure 1). Cutting cone activity grants bone healing without the formation of callus, intermediate fibrous, or cartilaginous tissue [14].

The cutting cone is a cellular structure that originates in the proximity of a neurovascular bundle and is generated by the propagation of the basic multicellular unit (BMU) [15]. In cortical bone, it has a distinctive morphology that resembles a cone with its apex pointing toward the direction of bone resorption. The 3D trabecular structure of cancellous bone makes complete visualization of the BMU within a cutting cone virtually impossible [16]. The cutting cone comprises a group of osteoclasts at the front of the cone, followed by osteoblasts, reversal cells, and secondary osteoclasts in the so-called reversal zone. As the cone advances through bone tissue, osteoclasts resorb the bone matrix, creating a tunnel-like cavity where osteoblast-mediated bone formation occurs via secondary osteons.

Finally, there is the closing zone, made up of osteocytes that finish the “filling” phase of the cone (sometimes referred to as a “filling cone”) [17].

*Secondary healing*, also called indirect healing, is another type of bone healing. This is generally much more common than primary healing, which is typical for fractures not managed by osteosynthesis and, therefore, without absolute stability. Secondary healing has four stages and proceeds from hematoma formation with subsequent inflammation, triggering cytokine production and cell migration [18]. The second stage is fibrocartilaginous remodeling of the hematoma and simultaneous subperiosteal woven bone deposition, resulting in *soft callus* formation [19]. In the third stage, the soft callus is remodeled into a bony callus through endochondral ossification, which in the final fourth stage, is remodeled into fully functional mature lamellar bone.

### 3.3. Conditions for Successful Bone Regeneration Focusing on Guided Bone Regeneration (GBR)

It is inevitable to emphasize the difference between **bone healing (repair)** and **bone regeneration** in the context of regeneration procedures, especially guided bone regeneration (GBR). These definitions are quite specific and, for the purposes of this review, were adopted from Misch’s Contemporary Implant Dentistry [10]. Bone healing, as mentioned earlier, is a process of restoring function and continuity of injured bone tissue, whereas bone regeneration encompasses multiple surgical procedures aimed at regrowing bone in areas where it was lost due to trauma, surgery, or a pathological process, or even in regions where it was never present before (sinus lifting or congenital defects including facial clefts). Therefore, it is a much more demanding procedure than bone repair [10].

Bone regeneration is supported by the biological principles of **osteogenesis**, **osteoinduction**, and **osteoconduction**. *Osteogenesis,* as a fundamental term in bone regeneration, includes all processes of new bone tissue formation. *Osteoinduction* refers to the processes of migration, proliferation, and differentiation of cells involved in bone formation. BMP (bone morphogenic protein) is currently the most researched and established osteoinductive factor. *Osteoconduction* expresses the ability of the material/scaffold to promote the adhesion of bone-forming cells and subsequent osteoid deposition on its surface.

Additionally, today the term *osteopromotion* may be used as it contributes to the enhancement of osteoinduction of the material, even though osteopromoters themselves do not have osteoinductive properties [10]. Examples of substances with osteoinductive properties are platelet-rich fibrin (PRF) and platelet-rich plasma (PRP).

Regarding the use of bioactive scaffolds implanted in bony defects, *osteointegration* has to be mentioned. The term osteointegration is used mainly in dental implantology and means ensuring mechanically stable direct contact between bone tissue and implanted material without intervening fibrous tissue [20].

Taken together, bone regeneration in terms of TERM is an extremely complex process, achievable through interactions of **cells**, signaling **molecules**, and a suitable **scaffold** in a favorable environment over a period of time [10,21].

Guided bone regeneration (GBR) was first described as a technique used in maxillofacial surgery to gain alveolar bone to allow favorable placement of dental implants. The original idea described the need for fast-proliferating cell exclusion from the osseous defect to provide uninterrupted bone healing [22]. The exclusion medium, the *membrane*, “guides” the newly formed bone into the desired area [23].

Wang et al. [23] defined four major doctrines (called PASS) for successful GBR, which are as follows: *Primary wound closure, Angiogenesis*, *Space maintenance,* and *blood clot Stability*. However, we can modify these basic principles based on current research findings. In the following paragraphs, ordered according to their significance (from our point of view), we advocate the importance of particular principles.

#### 3.3.1. Stabilization and Immobilization of Grafted Material

During initial fracture healing, a study demonstrated that more stable (rigid) fixation of bone fragments resulted in smaller callus formation with a reduced fibrous tissue component [24]. The importance of stabilization has also been proved in later stages of fracture healing, where the lower magnitude of rigidity resulted in the inhibition of callus stiffness and bridging [25]. In clinical practice, when performing the augmentation technique, to prevent the movement of the augmented or grafted material, it is usually necessary to use fixation screws, pins, or local tissue flaps to allow predictable healing [26,27].

The importance of graft immobilization can be explained by means of cutting cone progression. In primary healing, where absolute stability is requested, the bone can heal faster and the lamellar structure can be attained without any fibrocartilaginous intermediate. Otherwise, secondary healing with intramembranous and cartilaginous ossification takes place.

In bone regeneration techniques, usually, both principles occur [28]. However, their proportion depends on the technique and (grafted) material used, as well as local factors and the postoperative management of the patient.

The greater shear stress and subsequent prolonged healing may traumatize the fragile newly formed vessels, subsequently reducing the blood supply to the area where new bone is to be created. Proper angiogenesis is inevitable for cutting cone development and sufficient bone regeneration, as osteoclasts are descendants of bone marrow stem cells. Insufficient delivery of oxygen and nutrients results in osseous tissue of lower quality or even the formation of fibrous tissue only. This idea is supported by in vivo experiments where smaller interfragmentary movements resulted in the formation of a greater number of blood vessels compared to reduced vasculature, a higher proportion of fibrous tissue, and a lower degree of mineralization in areas of larger movements [29,30].

In the case of bone regeneration by augmentation procedures, it is, therefore, inevitable to immobilize the grafted material to allow for as fast and superior bone formation as possible.

#### 3.3.2. Volume Maintenance and Mechanical Support Provided by a Membrane

Sufficient space is inevitable for osteogenesis to ensure undisturbed proliferation of bone-forming cells and deposition of osteoid. This is especially the case in situations where larger defects are to be regenerated and the overlying soft tissue exerts substantial pressure over the treated area [23].

According to a recent review [31], successful vertical bone regeneration, which is generally considered more demanding than horizontal, was achieved with titanium-reinforced membranes. These membranes are conventionally perforated, so essentially, they cannot separate the bony defect from the overlying soft tissue completely. This is in contrast with the generally accepted primary goal of a membrane to achieve selective cell repopulation of bone defects and allow slow-working bone-forming cells to produce bone without any interruption from overlying fast-proliferating soft tissue cells. Still, they are generally considered to be the best choice for vertical GBR [32]. A recent study [33] showed that bone formation was significantly enhanced using a mechanically improved synthetic membrane that was adjusted to withstand 55 times higher bending force than a commercial product, again supporting the importance of space maintenance.

The occlusive properties of a membrane are also questioned by the concept of “transmembrane vascularization”, which promotes angiogenesis and improves the vascularity of the grafted area to achieve better regenerative outcomes. The partial fragmentation of the collagen membrane allows for so-called “secondary porosity”, which facilitates substantial vascularization and adds to the membrane’s functionality. Pore creation includes at least partially losing the shielding function of the membrane, yet it still yields superior results in bone formation [34,35].

These findings also support the conclusion that the function of a membrane as a volume maintainer is superior to its function as a barrier for non-osteogenic cell type exclusion [36].

#### 3.3.3. Membrane Compartment as a Promoter of Desired Molecular Events

The role of a membrane (either resorbable or non-resorbable) in GBR as only a passive mechanical barrier for rapidly proliferating cells has already been surpassed [37,38]. Applying a membrane over a bony defect creates a specific microenvironment that supports the healing process on a molecular basis [39]. Furthermore, membranes can be further artificially improved by adjusting their specific composition and structure to enhance bone regeneration in different applications.

According to Turri et al. [38], the presence of the collagen membrane over the bony defect resulted in various cell attraction into its scaffold with resulting expression of osteocalcin and the RANKL gene by those cells. These are important trigger molecules in the osteogenic process. In addition, the experimental membrane acted as a depository of BMP-2, FGF-2, TGF-beta, and VEGF produced by nested cells. These cytokines are known to participate in angiogenesis, migration, and differentiation of endothelial cells, fibroblasts, and osteoblasts during bone regeneration [14]. The underlying chemotactic mechanism of the collagen membrane is most likely caused by the up-regulation of two cell recruiting factors, i.e., CXCR4 (CXC chemokine receptor type 4) and MCP-1 (monocyte chemotactic protein-1) [38].

Taken together, these molecular and biological pathways resulted in surplus bone production in comparison to areas without any membrane treatment.

The membrane demonstrated the ability to promote high osteoclastic activity and concurrently recruit osteoprogenitor cells in the bone defect, which are crucial for bone remodeling, i.e., coupled bone resorption and subsequent bone formation [40]. This is again supported by the idea of cutting cones, where BMUs containing both osteoblasts and osteoclasts must work in close cooperation.

The osteoclastic activity itself was proposed in multiple studies as the main stimulus for engineered bone regeneration due to osteogenic substances released from the resorbed tissue as well as cytokines produced by osteoclasts themselves [41,42,43].

The ideal membrane according to the considerations mentioned above should be able to significantly promote a faster healing process together with an increase in the volume of bone produced [38]. Collagen, as an inherent body substance and the major component of bone tissue, seems to bioactively be the most promising material for membranes [39].

#### 3.3.4. Other Aspects

Regarding wound closure, the aim is to achieve primary tension-free closure whenever possible [23]. However, the clinical use of d-PTFE membranes has shown that localized membrane exposure does not have to inflict bone regeneration and does not result in graft failure [10,32,44].

Currently, various other tissue derivates (e.g., PRF and PRP), growth factors, cell-based therapies, gene therapies, and tissue engineering processes are being used or developed in order to further enhance, contribute to, and speed up the process of new bone formation. However, they are not the focus of this review.

## 4. Features of an “Ideal” GBR Membrane

Membranes used for GBR can be essentially divided into *resorbable* and *non-resorbable membranes* according to their ability to decompose in living tissue. In terms of membrane composition, constructs are defined as *polymer* or *non-polymer* membranes [45].

Each membrane has its own special advantages and disadvantages and is used in clinical practice accordingly. The aim of manufacturers in recent years has been to provide us with a membrane that is ***resorbable*** and ***functional***. The GBR membrane scaffold should allow loading with *antibacterial* and *bioactive factors* to reduce the failure rate caused by inflammation and promote osteogenesis as well. Furthermore, the *multilayer* structure is preferred to meet the different requirements of various contact surfaces, such as bone, submucosal tissue, or implant surfaces [45].

There are general characteristics that could be attributed to the “ideal” GBR membrane. Currently, none of the products on the market meet all of these features.

### 4.1. Biocompatibility

This is an inevitable characteristic of the scaffold for its safe use in TERM. Biocompatibility is the “ability of a biomaterial to perform its desired function, without eliciting any undesirable local or systemic effects on the recipient” [46]. Biocompatibility ensures the generation of non-toxic degradation products and the absence of adverse effects [47], i.e., cytotoxicity, immunogenicity, and genotoxicity, during the presence of the scaffold in a living organism.

### 4.2. Bioactivity

Bioactivity refers to the ability of the biomaterial to elicit a specific biological response in a host tissue [47]. The osteoconductive scaffold is supposed to interact with surrounding tissue and promote cell attachment, ingrowth, proliferation, and differentiation [10,45,48]. As mentioned earlier, the membrane has the potential to serve as a depository of various substances, i.e., growth factors and cytokines, which can *functionally* enhance bone regeneration by providing osteoinductive properties.

Furthermore, in the case of gradual scaffold degradation in vivo, membrane degradation products, including stored bioactive substances, have the potential to be released continuously to improve the “guiding” of bone regeneration [38].

### 4.3. Mechanical Strength (Including Tensile Strength)

As noted earlier, space maintenance is one of the most important conditions of GBR, which is a slow and complicated process that requires absolute stability and minimal movement of the graft [45]. Sufficient mechanical strength is, therefore, crucial for a functional membrane. Moreover, it enables us to manipulate the material safely and decreases the risk of tearing. On the other hand, the stiffness of the material cannot prevent a surgeon from comfortably handling it to ensure fast and reliable procedure completion.

The tensile strength cannot be omitted as part of material mechanical endurance, as we know that the majority of GBR augmentation techniques, including the Istvan Urban “sausage” technique, require significant stretching of the collagen membrane over the bony defect and its fixation in a stretched state in order to prevent the graft from moving [49].

The mechanical properties of the membrane improve with increasing membrane thickness, further increasing the space maintenance ability of the scaffold [45,50]. Furthermore, the increase in membrane thickness reduces soft tissue ingrowth into defects [39]. On the other hand, excessive thickness limits the primary closure of the defect and worsens the handling of the product.

### 4.4. Biodegradability and Remodelability

As stated previously, the ideal GBR membrane is *resorbable* and *functional.* However, especially in larger bone defects, this cannot be fulfilled as resorbable membranes do not provide sufficient mechanical support for the regeneration process in comparison to non-resorbable counterparts, which remain the gold standard in complicated and large augmentation procedures despite the need for second surgery and increased morbidity of the patient. However, recently emerged novel non-polymeric resorbable membranes manufactured from magnesium (Mg) or zinc (Zn) are showing promising results. In vivo, Mg and Zn membranes have been shown to be osteogenic and have a bioactivity superior to that of practically inert titanium mesh [51,52,53]. Due to the coupling advantages of resorbable and non-resorbable materials, such as enhanced mechanical stability without the need for removal, these membranes seem like promising alternatives.

In the ideal case, the resorption rate of the scaffold should coincide with the rate or regeneration of bone tissue [10] and, therefore, the programmed or adjustable degradation rate is desired to meet specific needs in different cases and locations. The scaffold residuum, which is not degraded, is, in the ideal case, incorporated into newly formed tissue. Only a few studies have demonstrated the **osteogenic** properties of collagen membranes in vivo. The porcine collagen membrane was shown to induce differentiation of osteoblasts, transform gradually into bone tissue while resorbed, and create the bony layer over the artificially created bony defect [54,55,56].

### 4.5. Antibacterial Properties

Bacterial contamination of the bony defect may result in decreased or completely failed bone regeneration. Additionally, bacterial ingrowth into degradable membranes, together with their metabolic products, causes faster scaffold degradation and, therefore, cancels its fundamental function of space maintenance [57].

The membrane can block bacteria entry through a variety of mechanisms. Structure adjustments (e.g., decrease in porosity) as well as the incorporation of antimicrobial substances into the material may help to prevent barrier contamination.

### 4.6. Optimal Porosity

Scaffold porosity, pore size, interconnectivity, and tortuosity are some of the most important parameters in bone tissue engineering [58,59,60]. The porous structure and surface features of the membrane create an environment for cell attachment, thus enhancing the bioactivity of the material and are also inevitable for vascularization [38,61].

The research about the optimal pore size of the GBR membrane is inconclusive but highlights the importance of the entire scaffold complexity and interconnectivity of the pores, as each pore size has specific benefits but also drawbacks. It is also necessary to mention the importance of the *open porosity* of the scaffold as well as the pore connection within the scaffold and with the external environment, which are crucial for cellular processes including angiogenesis. This contrasts with so-called *closed porosity,* where a scaffold’s internal porous structure remains isolated and does not communicate with its surroundings [58].

**Macroporous structure** (>100 μm) is superior in host bone tissue integration, angiogenesis, and vascularization [61]; cell proliferation and differentiation; osteogenetic cytokine secretion [62] and permeability rate, resulting in a higher osteogenetic potential of the grafted site and subsequent osteogenesis [63]. This may be reasonably explained by the diameters of BMU cells, which have to “settle” within the scaffold to induce bone formation. Osteoblasts range from 10 to 30 μm and osteoclasts from 100 to 300 μm. In most experiments, the ideal pore size of the scaffold for new bone formation, with reasonable mechanical endurance, is around 300 μm and not larger than 400 μm [63,64,65].

**Microporous scaffolds** (<100 μm) are significantly superior in the resistance to compression stress [66]. A more compacted structure also results in a slower degradation rate and, therefore, a slower and more controlled release of incorporated substances. The microporous scaffold disposes of a larger surface area and, therefore, better adherence of the cells [61,67,68]. It is also superior as an antibacterial barrier and, even in the case of wound dehiscence, a scaffold with sufficiently small pores could maintain osteogenesis uninterrupted [44]. However, its disadvantages include limited nutrient transport, the risk of pore obliteration during swelling of the membrane, the loss of pore interconnectivity and, therefore, worsened vascularization, cell proliferation, and activity.

### 4.7. Water Absorption Capacity

The cellular and molecular processes involved in bone regeneration and angiogenesis require a moist environment. This also includes the need for scaffold hydration [69]. In addition to enabling cellular function, the membrane must allow oxygen, water, and nutrient inflow and waste removal. Therefore, scaffold permeability plays a crucial role in the regeneration process, with higher membrane permeability resulting in increased bone growth [63,70].

Collagen scaffolds are easy to hydrate and prone to swelling due to their hydrophilic structure [71]. According to Suchý et al. [58], collagen scaffold changes minimally with hydration. This results in no effect on the porosity of the scaffold, the size of the pores, or the persistence of the permeability of the scaffold.

However, hydration influences the mechanical properties of the scaffold, resulting in a statistically significant decrease in the elastic modulus and compressive strength of the material [58].

This must be considered because membranes are used in a hydrated setting in clinical practice.

## 5. Fish-Derived Collagen

Collagen, with more than 20 of its subtypes, is the most abundant protein in the human body, the predominant component of the extracellular matrix, and the main organic part of bone tissue [8]. Type I, the main type of collagen [72], produced by osteocytes, osteoblasts, and fibroblasts, has an integral role in maintaining bone structure, structural support of tissues, and regenerative processes [10,32,73]. Collagen favors the coagulation process and cell adhesion to its surface, with subsequent proliferation and differentiation [2,10,38]. Compared to non-resorbable membranes, collagen membranes have a lower incidence of membrane exposure [74,75] and because of their restorability, there is no need for a second surgery to remove them. When combined, this increases patient compliance and reduces their morbidity. Moreover, its degradation mimics physiological tissue turnover and produces non-toxic degradation products [76]. Therefore, it is not surprising that it has proven functionality, biocompatibility, and convenience, and is popular in bone regenerating procedures, becoming the most widely used membrane in conventional GBR [77].

Collagen I is a fibrous protein made up of two α1 chains and one α2 chain, forming a triple-helix (three-helix) structure [78]. It belongs to hydrophilic proteins with a typically repeating sequence of amino acids, Gly–X–Y, where X is commonly occupied by proline (Pro) and Y by hydroxyproline (Hyp). The amino acid content affects the mechanical properties of collagen and influences its cross-linking ability (described later in the text) [78]. Pro and Hyp stabilize the three-helix structure; therefore, the integrity of collagen, together with lysine (Lys) and hydroxylysine (Hyl), is necessary for the formation of intramolecular and intermolecular bonds [78,79]. To produce the desired *functional, biologically active* scaffold, the three-helix structure of collagen must be preserved [59].

The collagen source was shown to be an important factor influencing the bone regeneration process and the degradation rate of the scaffold used [32].

Currently, the most common xenograft source of collagen for TERM scaffold production is bovine and porcine type I and III collagen from the skin, the tendon, and the pericardium, which is homologous to human collagen by structure and amino acid content. These types of collagen have shown favorable outcomes in experimental and clinical practice and have proven to be a reliable option as GBR membrane material.

However, the use of collagen in TERM, including GBR, is somewhat limited, especially in the management of larger defects, due to its poor mechanical properties and relatively fast enzymatic degradation in the case of untreated “pure” collagen [80]. Furthermore, collagen scaffolds alone, due to their low structural stability, are not sufficient for space maintenance, which is their main limitation for use in clinical settings [81].

Natural polymers, including collagen scaffolds, could initiate a strong immunogenic reaction and require complex purification to decrease the rate of disease transmission and host–body reactions [32].

In recent years, there has been emerging research on waste products from nonmammalian organisms, that is, marine and freshwater fish, jellyfish, squid, and shark, as an alternative source of collagen [1]. Their benefits include virtually no risk of zoonoses due to the large ontogenetic distance to mammals [82], more economically pleasing solutions, and the absence of religious constraints [32,83,84,85,86]. Furthermore, freshwater fish collagen is less immunogenic than its mammalian counterpart; moreover, it has shown its effectiveness in direct cell adhesion and subsequent cell differentiation, as well as excellent biocompatibility [2,87,88].

However, fish and mammal collagens differ in amino acid content [78], especially Pro and Hyp, which are amino acids responsible for the structural stability of collagen and thus its bioactive function, as mentioned above. Fish collagen, due to the lower content of Pro and Hyp in their native form, has inferior mechanical properties compared to mammalian collagens [89].

Another problem is the heat stability of the particular collagen type and its denaturation temperature, above which the structure of the collagen molecule is no longer preserved, and durability and scaffold function cannot be granted [90,91]. Fish-derived collagen generally has a lower denaturation temperature than mammalian products [3]. However, the specific denaturation temperature of a particular fish varies depending on its habitat and its normal body temperature. Collagen obtained from freshwater fish typically has a higher denaturation temperature than marine collagen, giving it an advantage in human medicine use [2,92].

### 5.1. Improving Fish Collagen Characteristics

Fish-derived collagen membrane is limited by its low stiffness, melting temperature, fast degradation rate, and less reproducible properties compared to mammal collagen [6,32,88]. These drawbacks are a potential cause of GBR failure in clinical practice [93].

Physicochemical modification of the scaffold is, therefore, inevitable for the effective use of collagen in bone regeneration procedures. Yang et al. [45] suggested four ways of improving collagen characteristics, i.e., *cross-linking, electrospinning, multilayer superposition*, and the *addition of different biological materials*.

#### 5.1.1. Cross-Linking

Cross-links are covalent bonds within collagen fibrils that are responsible for their mechanical properties. In all species, lysyl oxidase forms cross-links in non-helical telopeptide regions by converting Lys and Hyl residues to their corresponding aldehydes [78,94]. However, the naturally formed cross-links found in fish collagen are acid-labile [78] with the resultant mechanically insufficient collagen properties.

Cross-linking is an effective strategy to increase the tensile strength, stiffness, compressive modulus, resistance to collagenase, and durability of collagen scaffolds [32,95,96]. Through this process, bonds are created between and within collagen molecules [97]. Furthermore, cross-linked collagen shows less antigenicity [78]. However, a high degree of collagen cross-linking is associated with a higher exposure rate of the scaffold material and a potential foreign body reaction [32].

There are several methods for cross-linking, according to the agent used: *chemical*, *enzymatic*, and *physical*.

**Chemical cross-linking** is the most common technique for processing collagen scaffolds. The agents used can be divided into natural and traditional chemical solutions [32]. The most frequently cited natural representative is *genipin*, which produces membranes with great biocompatibility and mechanical properties, albeit at a high cost. Other natural agents in the current research are *proanthocyanidins* and *epigallocatechin-3-gallate*. Traditional chemical cross-linkers are much more common, with the following representatives: aldehydes, epoxides, isocyanates, and imidates, etc., which are effective in enhancing collagen mechanical properties [78]. From these, glutaraldehyde is generally well known and still relatively widely used because of its low cost and efficiency. However, together with many other agents in this category (e.g., glyoxal, formaldehyde, hexamethylene diisocyanate, etc.), it poses the risk of local tissue inflammation and calcification and has been reported to be cytotoxic [78].

In contrast to other traditional chemical cross-linkers, all residuals from the N-(3-dimethylaminopropyl)-N-ethylcarbodiimide hydrochloride (EDC) and N-hydroxysuccinimide (NHS) cross-linking process are water-soluble and can be easily washed out by water; therefore; this cross-linker represents no cytotoxicity. EDC-NHS cross-linked collagen has superior mechanical features, a more compact structure, and excellent biocompatibility in comparison to commercial non-crosslinked collagen membranes (BioGide^®^, Gestlich Pharma AG, Wolhusen, Switzerland) [59,98,99,100]. When used in fish-derived collagen processing, the triple-helix structure remained well preserved; therefore, the EDC-NHS cross-linked scaffold remains bioactive [59]. Cross-linking with EDC-NHS significantly improves biostability and degradation kinetics by varying the cross-linked density [2].

**Enzymatic cross-linking is** most commonly performed by *transglutaminase* and produces scaffolds with favorable biocompatibility and biomimetics, however, with much lower mechanical stiffness and relatively high cost [32].

**Physical cross-linking** is performed via *UV* (ultraviolet) or *DHT* (dehydrothermal) treatment. Collagen cross-linked by UV irradiation partially denatures [101]. In addition, the mechanical endurance and stiffness of UV cross-linked material is not as good as in the case of chemical cross-linking. To increase cross-link density, UV light is used in combination with photo-activatable reagents (e.g., riboflavin) to produce links within and between collagen molecules [101]. DHT curing of collagen produces a scaffold with higher tensile strength, but the process is considerably exhaustive and time-consuming [102].

#### 5.1.2. Electrospinning

The electrostatic spinning of collagen creates nanofibers from a polymer solution, which significantly increases surface area and mechanical strength, and improves cell adhesion and bioactivity. However, spinning can be technically demanding, and it is especially difficult to maintain a balance between successful scaffold formation and the preservation of the collagen structure [78].

#### 5.1.3. Multilayer Superposition

Combining several layers of collagen can create a scaffold with enhanced mechanical properties. Furthermore, membrane surfaces facing specific tissues (e.g., bone or connective tissue) can be adjusted to provide as natural an environment for relevant cells as possible. The adjustment of layers may be either mechanical, e.g., by changing pore size and shape, or chemical by impregnating a particular layer with desired substance/stem cells/cytokines, etc. The goal is to create an individualized surface for a specific tissue that is in contact with a particular scaffold’s surface.

#### 5.1.4. The Addition of Different Biological Materials

Modification of the collagenous scaffold has the capacity to alter its bioactivity, which is the way to enhance cell migration, proliferation, and differentiation within the scaffold [6].

## 6. Introducing Novel Carp Collagen Sponge

The new carp-derived collagen sponge (Figure 2) is manufactured from the skin of the European carp *Cyprinus carpio* according to Patent No. CZ308862B6 [5].

In brief, the “*collagen sponge*” is prepared by mixing the weighed amount of collagen I with deionized water to allow collagen swelling. After homogenization, the resulting dispersion is frozen and subsequently lyophilized. Next, chemical cross-linking via EDC and NHS is performed to ensure sponge stability. Thereafter, the sponge is refrozen, lyophilized, and eventually optionally impregnated with desired substances (e.g., antibiotics, antivirals, antimicrobials, etc.) [5]. Due to processing methods, the hydrated collagen sponge has elastic memory, is strong when bending, and has favorable handling properties, including a reduced risk of tear [5].

The extracted carp collagen has a heat-shrink temperature of approximately 20 °C, which can increase with cross-linking to 43 °C [103]. This provides stability for the carp-derived scaffold use under normal human body temperature conditions. Additionally, lyophilization (freeze drying) is an ideal method for creating scaffolds with numerous pores to promote vascularization and osteogenic cell proliferation [104].

Collagen sponge and its derivatives (described later in the text) can be sterilized and used safely during surgical procedures. Exposition to a nominal dose of 25 kGy does not inflict any damage on the material and preserves its internal structure [5,105].

The porosity as well as the degradation rate of the novel collagen sponge can be programmed and set during the manufacturing process and subsequent experiments for verification. Porosity can be changed either by collagen concentration or by changing the freezing temperature before final lyophilization. The more concentrated the collagen dispersion, the smaller the pore size. The lower the freezing temperature, the smaller the pore size [5]. The scaffold degradation rate is altered by the length of the cross-linking process and the temperature during that process. The longer the cross-linking process, the slower the degradation rate that the scaffold exhibits. Subsequently, the controlled degradation time of the scaffold dictates the release time of the active substance(s) stored in the sponge [5].

This kind of collagen processing allows us to overcome the potential disadvantages of fish-derived collagen, that is, denaturation temperature, scaffold stability, degradation rate, and promote the bioactivity of the scaffold as well.

The patented carp collagen sponge derivative called the “*sandwich collagen sponge*”, consists of a highly porous carp collagen sponge in the peripheral layers and a low porous collagen sponge in the central part (core) (Figure 3). It is manufactured from collagen dispersions with various concentrations. The pore size in the core is in the range of 20–80 µm and in the periphery 50–200 µm [5]. This architecture creates a scaffold with advantages from both the microporous and macroporous structures described above. Hartinger et al. determined in vitro that the degree of closed porosity of the sponge was very low, that is, between 2 × 10^−5^ and 4 × 10^−3^% [106]. This value indicates a minimal proportion of non-interconnected pores and predominant open porosity, which is crucial for the vascularization of the scaffold as well as cellular processes.

The next derivative of the carp collagen sponge is a “*composite collagen sponge*” comprising at least two layers, that is, a nanofiber collagen layer prepared by electrostatic spinning and a porous collagen sponge, as described earlier. These two layers alternate, creating a composite of several layers (Figure 4). This composite material is advantageous, particularly in cases where increased surface area is needed, such as in cell adhesion. Moreover, a greater surface area allows a greater loading of active substances and their subsequent release at higher concentrations [5].

Sandwich as well as composite scaffolds are stable and layers do not delaminate from each other in vivo and in vitro [5,106]. All novel products are highly hydrophilic with a highly porous *collagen sponge* manufactured from 1% *w*/*w* collagen dispersion absorbing fluid up to 30 times its weight. A low-porosity sponge of 5% *w*/*w* can absorb fluid up to 15 times its weight [5]. This secures a favorable environment with sufficient water permeability for cellular processes. Hartinger et al. proved that a hydrated *sandwich sponge* is significantly stiffer than a homogeneous sponge and is capable of withstanding higher loads [107].

The composite collagen sponge and the low-porosity homogeneous sponge are mechanically superior (Figure 5) and, therefore, could provide better protection of the bone-grafted area and resist pressure from the overlying soft tissue, even in cases of greater bone defects.

### 6.1. Biocompatibility, Safety, Allergenicity, and Immunogenic Potential of a Novel Carp Collagen Sponge

The success of TERM is directly dependent on the construct used [108]. The main cause of graft or implant rejection is the negative immune response of a particular host [1]. However, the extent of rejection and subsequent failure in tissue regeneration also depends on the properties of the graft/implant, that is, material, amount or volume delivered, quality, purification, and its artificial modification.

Scaffolds created by recombinant procedures are practically non-immunogenic, but their significant cost and prolonged manufacturing make them hardly available for clinical practice. Accordingly, xenogeneic and allogenic scaffolds are preferred.

Collagen is generally regarded as a biocompatible, biodegradable, and non-toxic material that can be obtained from various animal sources (e.g., skin, bone, tendons, scales, etc.), which often ends up as waste material from industrial sectors [78].

Mammal collagens are highly homologous in most of their domains to human collagen; however, there are significant differences in telopeptide regions [72] and after their removal by telopeptidase, there is a 3–4% risk of allergic reaction in the population [109]. In addition, there is a risk of transmissible diseases, which are a potential problem with bovine and porcine-derived biomaterials, such as bovine spongiform encephalopathy, transmissible spongiform encephalopathy, and foot and mouth disease contamination [110,111,112].

Fish-derived collagen, which does not pose a risk of disease transmission or allergic reactions, therefore, constitutes an alternative safer construct for TERM. Marine collagen (fish, jellyfish, corals, and algae) proved to be biocompatible [21,113,114] and jellyfish collagen, the most studied, was shown to provide even higher cell viability than bovine collagen [113].

Collagen derived from freshwater fish scales (*Catla catla* and *Labeo rohita*) did not show considerable signs of cytotoxicity or immunogenicity in vitro [2]. In addition, in vivo mouse models injected with this fish-derived collagen did not demonstrate signs of systemic toxicity. Favorable results regarding the safety of fish collagen were found when testing the collagen scaffold derived from tilapia in vitro and in vivo [115].

Lambert et al. [3] evaluated the immunogenicity of carp (*Cyprinus carpio*)-derived collagen in vivo and showed only a mild or even slightly decreased immune reaction to a new carp collagen sponge and regarded it as comparable to or even better than commonly used bovine collagen.

Taken together, from an immunologic point of view, there is scientific evidence that the freshwater-fish-derived collagen seems at least as appropriate or even superior for TERM use as bovine and porcine collagen.

### 6.2. Medical Applications of Novel Carp Collagen Sponge

Several in vitro and in vivo studies explored the potential of this novel material with promising results.

#### 6.2.1. Wound Dressing and Drug Delivery System

Wound dressings impregnated with drugs, especially antimicrobials, are advantageous, particularly in the case of aminoglycosides, vancomycin, and nitrofurantoin, which have poor distribution in targeted tissues. The local delivery system could overcome this problem and thus prevent the use of reserve antibiotics such as linezolid or ceftaroline, which should only be used as a last resort.

A collagen sponge manufactured from *Cyprinus carpio* was used for the first time as a wound dressing by Lukac et al. [116]. It was applied in a rat model to treat infected wounds and proved effective in the elimination of *Pseudomonas aeruginosa* colonies, with greater effectiveness compared to intramuscular application.

Hartinger et al. [106] proposed the use of this sponge impregnated with vancomycin as a wound dressing to prevent surgical site infection or to treat already infected wounds. The collagen sponge maintained its stability and, over 7 days, gradually delivered the antimicrobial agent to the implantation site, resulting in a significantly reduced number of MRSA colony-forming units (CFUs) in the rat model.

Another study [107] demonstrated consistent rifampin release from the “sandwich collagen sponge” and a longer preservation of *Streptococcus* MIC levels in surrounding tissues and lower levels in the systemic circulation; therefore, making it a more effective and safer alternative to systemic administration of the antibiotic.

The application of such an effective dressing may be especially beneficial in poorly perfused wounds (e.g., chronic ulcers), where much lower drug penetration from the systemic circulation can be anticipated [106,116]. Furthermore, with local administration, the systemic toxic side effects can be partially eliminated as the systemic concentration does not reach the level of local concentrations. Moreover, such a dressing remains bioactive and promotes tissue healing due to its inherent collagen scaffold characteristics even after complete drug release [106].

#### 6.2.2. Vascular Graft Coating

In cases where autologous vascular grafts are not available or are not sufficient, prosthetic grafts act as a substitute. Carp collagen sponge was used in vivo to test a novel three-layer composite vascular graft [103], which consists of a central polyester mesh covered from both sides by the carp collagen sponge. Two variants were created, one with a lower fat content and, thus, a higher relative carp collagen content. The graft coated with a sponge with a higher collagen content and less fat was shown to be more efficient and allowed re-epithelization from the periphery while concurrently maintaining graft shape stability.

## 7. Discussion

### 7.1. Potential Use of Carp Collagen Sponge in Bone Healing and Regeneration

#### 7.1.1. Bone Defect Management, including Extraction Socket Preservation

Bone tissue has very limited spontaneous healing capacity, and thus is not surprisingly one of the main fields of interest in TERM [117]. To adequately support such a defect and successfully manage bone loss, it is inevitable to develop such a bone graft that is capable of all biological conditions for bone regeneration, namely osteogenesis, osteoinduction, and osteoconduction.

The new carp collagen sponge provides osteoconductive properties by itself and, due to its inherent structure, can be easily modified with other potent substances, such as hydroxyapatite, growth factors, Mg and Sr ions, and MSCs, which provide osteogenic and osteoinductive properties [118,119,120,121].

Marine and freshwater fish collagen has already been tested in hard tissue applications [100] with promising in vitro results suggesting its osteogenic potential, in particular, when modified by cross-linking and impregnated with calcium phosphate to enhance its mechanical properties and prolong its disintegration [88,117]. Thanks to the internal structure of carp collagen sponge and its derivatives, multilayered fashion, adjustable porosity, and degradation rate via cross-linking, we propose its use in managing bony defects as a scaffold providing osteogenetic, osteoconductive, and osteoinductive properties. Regrettably, the number of in vivo studies is very limited; therefore, there is an urgent need for further trials of such a potential material.

The residual fat content in the novel carp collagen sponge allows vitamin D to be instilled inside the scaffold. This is an interesting suggestion, as vitamin D is known for its role in bone metabolism. Currently, increasing attention to its role has led to the discovery of D hypovitaminosis as a common syndrome in the human population [122]. A recent review that analyzed local vitamin D administration by bone graft showed improved osteointegration of dental implants and increased bone formation in animal models [123]. Unfortunately, research on this topic is scarce, and further trials are needed to support the benefits of local vitamin D administration to introduce it into clinical practice.

As mentioned earlier in this review, blood clot stabilization has been known for many years to be one of the bone regeneration principles. Application of such a collagenous scaffold into the extraction socket is beneficial not only in means of mechanical support but also as an active structure for the nidation of cells and molecular processes. If proven effective in bone regeneration, the novel carp collagen sponge could be used as socket preservation filling to avoid subsequent augmentation in cases where immediate implantation is not possible (e.g., periodontal infection, the economic situation of the patient, etc.), therefore reducing the likelihood of bony ridge atrophy. The question arises about the mechanical stability of such scaffolds in bony defects, especially of larger size. This has to be tested in vivo and compared with widely used autologous, allogenous, and xenogeneic bone grafts.

#### 7.1.2. Guided Bone Regeneration Procedures—Membrane

As previously stated, the membrane has a crucial role in successful GBR. Its biocompatibility, favorable structure for cell adherence, proliferation, high absorption ability, and potential enhancement with potent substances (see paragraph above) could provide a favorable microenvironment for predictable bone regeneration. Hadzik et al. [124] already demonstrated the successful use of fish-derived collagen membranes in GBR in a rat model.

This novel carp collagen sponge, due to its porous structure, may be impregnated with various substances, i.e., antibiotics, chlorhexidine, or other antimicrobials, metal ions, enzymes, etc., to prevent subsequent infection of this scaffold and impaired bone regeneration.

Commercial collagen membranes are used primarily for horizontal alveolar ridge restoration as they do not possess enough mechanical strength to support vertical defects. Due to the superior resistance of the “sandwich collagen sponge” to compression stress, there might be a possibility of a successful result even in small vertical restorations. However, these suggestions need to be further researched and verified in vivo.

#### 7.1.3. Antibiotic Delivery System in the Management of Osteomyelitis/Osteonecrosis

The effectiveness of the novel carp collagen sponge in the delivery of antibiotics to soft tissue wounds was already proven [106,116].

The drug’s delivery ability could also be particularly beneficial in bone infection treatment. Due to insufficient blood supply to the infected or necrotic bone, the penetration of antibiotics from the circulation system could be insufficient; therefore, local antimicrobial therapy via scaffold implantation seems reasonable [125].

In several aspects, local antibiotic therapy appears to be superior to systemic therapy in osteomyelitis management. First, it reduces the risk and side effects of systemic antibiotic administration; second, its safety has been proven in the treatment of soft tissue and joint infections; and third, it is considerably less expensive than systemic therapy in outpatient and inpatient settings [125,126].

Furthermore, antibiotics in the collagen membrane have been proven to slow collagen degradation [45] and, therefore, supply the targeted tissue for a longer period of time. The cross-linked drug carrier can even be adjusted to be pH-responsive and provide controlled substance release [127,128].

In the case of the use of degradable material, such as collagen scaffold, a second surgery for delivery system removal would be avoided. The carp collagen sponge and its derivatives are prone to epithelization, allowing the secondary closure of the osteonecrotic defect without the need for surgical management [106,116,125]. On the other hand, the question arises whether the one-time antimicrobial delivery would be sufficient for the management of osteomyelitis and osteonecrosis, which often take a prolonged course and require long-term antibiotic administration.

#### 7.1.4. Soft Tissue Augmentation Procedures

While performing hard tissue augmentation in maxillofacial procedures, there is often a need to augment soft tissues as well. In practice, the surgeon usually must use several products in staged surgery to gain both.

Currently, only AlloDerm GBR (costly and not easily accessible) has been statistically proven to act as a barrier membrane and increase the volume of the overlying mucosal layer at the same time [129].

We propose the carp collagen sponge, especially its “sandwich” derivative, for use in one-stage soft and hard tissue regeneration. With sufficient thickness and a multilayered structure with different porosities, it could provide mechanically and biologically active support for bone regeneration as well as increase soft tissue thickness. Previous studies in vivo showed promising results on infected wound management and re-epithelization of carp collagen sponge [106,116]; therefore, eventual dehiscence and bacterial colonization could be successfully controlled.

#### 7.1.5. Other Uses

Collagen, thanks to its biocompatibility and characteristics, is used in various fields of regenerative medicine, as well as cosmetics and the food industry. The novel carp-derived porous scaffold might be successfully applied in dural repair, articular cartilage regeneration procedures, peripheral nerve repair, etc. These and other clinical applications are potential subjects for further research.

## 8. Conclusions and Future Directions

In this review, we summarized the well-known principles of bone regeneration and advocated for the potential use of collagen materials in that process.

From a clinical and evidence-based point of view, the most important factors for successful bone formation in the desired places are space maintenance and the immobility of the grafted material. According to current research, the membrane is also increasingly important in GBR as an active biological medium for remodeling and the osteogenic process, rather than as a passive barrier to prevent soft tissue cell ingrowth.

The novel European carp-derived collagen sponge can fulfill (according to in vitro studies) the mechanical requirements even better than some of the commercial collagen products used in today’s clinical practice. It is also biocompatible, and its degradation time and internal structure can be adjusted via cross-linking, according to specific requirements.

In addition, this advantageous scaffold may be impregnated with various substances or cells to enhance its healing properties.

All the advantages of novel carp collagen mentioned above advocate for its use as a promising material in bone regeneration methods, especially GBR. The subject of our following studies will be the monitoring of in vivo degradability, as well as the local tissue reaction to the prototype membrane manufactured from this advantageous fish collagen scaffold, to determine its potential use and predictability in bone tissue regeneration.

## Figures and Tables

**Figure 1 jfb-15-00150-f001:**
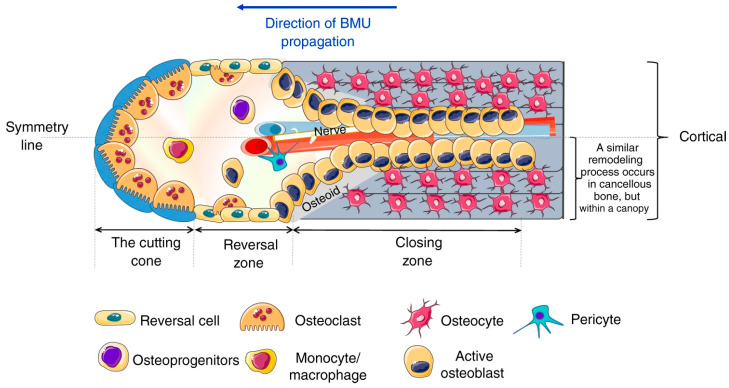
The cutting cone structure. Adopted from Bolamperti, S.; Villa, I.; Rubinacci, A. Bone remodeling: an operational process ensuring survival and bone mechanical competence. *Bone Res.* 2022, 10, 48. https://doi.org/10.1038/s41413-022-00219-8 [15]. Retrieved from: https://www.nature.com/articles/s41413-022-00219-8/figures/1, accessed on 29 March 2024. Note: the canopy is a layer of flat osteoblastic cells, which separates the marrow cavity from the resorptive and formative surfaces on the cancellous bone.

**Figure 2 jfb-15-00150-f002:**
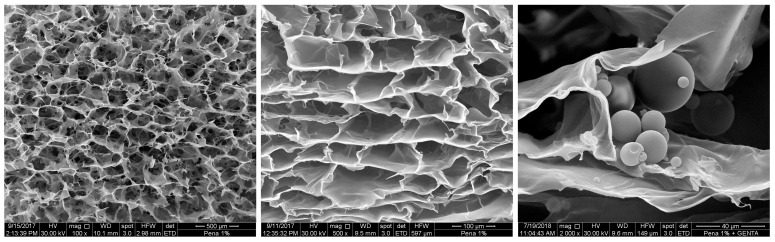
Simple collagen foam prepared by lyophilization of 1 wt% dispersion of type I collagen (skin, *Cyprinus carpio*). Left: mag. 100×, bar 500 μm; mag. 500×, bar 100 μm. Right: collagen foam after incorporation of the antibiotic gentamicin. Retrieved from: Patent No. CZ308862B6 [5]. Reproduced with permission from the authors (permission obtained 6 May 2024).

**Figure 3 jfb-15-00150-f003:**
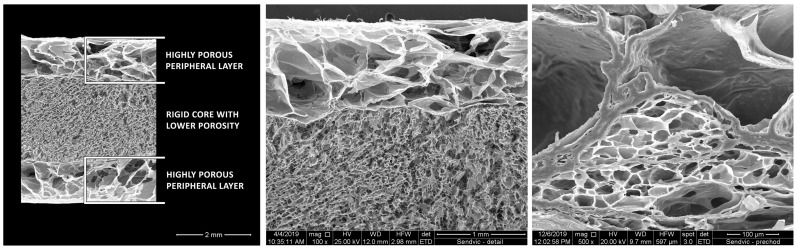
Sandwich collagen (type I, skin, *Cyprinus carpio*) foam prepared by sequential lyophilization of two differently concentrated collagen dispersions. The sandwich is composed of a rigid core with low porosity (5 wt% of the collagen dispersion), and the peripheral parts form highly porous layers (1 wt%). Left: macro image of the internal structure (bar 2 mm); detail of the cross-section of the interface between two differently porous layers (mag. 100×, bar 1 mm; mag. 500×, bar 100 μm). Retrieved from: Patent No. CZ308862B6 [5]. Reproduced with permission from the authors (permission obtained 6 May 2024).

**Figure 4 jfb-15-00150-f004:**
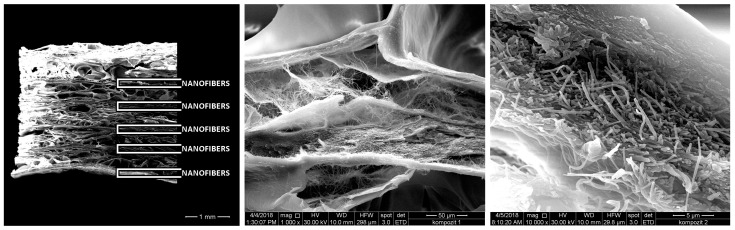
Composite collagen foam consisting of layers of collagen nanofibers and porous collagen (type I, skin, *Cyprinus carpio*) matrix. Left: macro image of the internal structure (bar 1 mm); detail of the nanofibrous layer in the porous foam (mag. 1000×, bar 50 μm); and detail of the cross-section of the nanofibrous layer (mag. 10,000×, bar 5 μm). Retrieved from Patent No. CZ308862B6 [5]. Reproduced with permission from the authors (permission obtained 6 May 2024).

**Figure 5 jfb-15-00150-f005:**
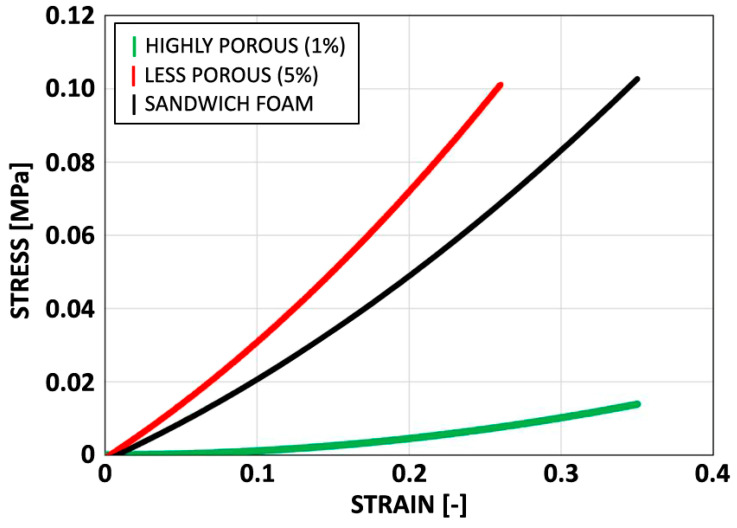
Stress–strain characteristics of the individual parts of a sandwich collagen foam obtained from monotonic tensile tests. Highly porous foam (prepared from 1 wt% collagen dispersion, green) is compliant, while less porous foam has a higher stiffness (5 wt% collagen dispersion, red). By combining both layers to form a composite material (or so-called sandwich), the resulting mechanical properties can be optimized (black). Retrieved from: Patent No. CZ308862B6 [5]. Reproduced with permission from the authors (permission obtained 6 May 2024).

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
