# Peer review of "Enhancing Guided Bone Regeneration with a Novel Carp Collagen Scaffold: Principles and Applications"

_jfb, 2024, doi:10.3390/jfb15060150_

Round 1
Reviewer 1 Report
Comments and Suggestions for Authors Please indicate the number of articles considered for this review Please indicate the inclusion and the exclusions criteria Many procedures and materials are presented...however, can you evaluate them comparatively based on some tests/references ? In the conclusion part "In this review, we have summarized the well-known principles of bone regeneration and advocated for the potential use of collagen materials in that process." - can you suggest one or more procedures with one or more materials options that will have the best results in this moments?Author Response
Please see the attachment

Reviewer 2 Report
Comments and Suggestions for Authors
The review by Bujda et. al, provides a nice overview on bone regeneration and application of fish-derived collagen. However, the structure needs some improvements, and some sections need some editing.
Please find below a list of points for revision:
· page 2: please delete the heading literature review
· page 2: 2.1. edit Bone regeneration
· page 4, 2.1.3: pleas edit as “conditions for bone regeneration.”
The structure is a bit confusing. There are a lot of small separate paragraphs that can be grouped together and summarized. For example, section 2.2.1 Biocompatibility can be grouped with bioactivity, biodegradability and remodelability.
Section 4. page7 should be incorporated in other sections.
The need for crosslinking collagen can be introduced at the end of section “Fish-derived collagen” and section 2.4 can be deleted.
The section chemical crosslinking should be revised, the information and classification are not entirely correct.
Collagen-based materials can be obtained via covalent or physical crosslinking. The main difference is that physical crosslinking can be reversible; crosslinking via EDC, genipin, enzymes, irradiation all result in covalent crosslinking. Hence, the classification is not correct.
Further, it should be mentioned that irradiation is usually performed in presence of photoinitiatiors (there is a lot of literature about this) and in some cases collagen is functionalized to present functional groups that can be used for photocrosslinking. All these information should be provided.
The section about electrospinning should be either expanded or removed. It is not clear why it is in the section of the crosslinking. Electrospinning is not a crosslinking method.
Same for section 2.4.4. the addition of different biological materials, or it is expanded or deleted.
In general, information can be summarized in a more concise manner so that the review becomes easier to follow.
Comments on the Quality of English Language
minor edit of English language required.
Reviewer 3 Report
Comments and Suggestions for Authors
jfb-3024390
Title: Enhancing Guided Bone Regeneration with a Novel Carp Collagen Scaffold: Principles and Applications
Overview
This is an interesting article that addresses, at the same time, aspects of a review on Guided Bone Regeneration (GBR), but also presents a scaffold obtained from collagen extracted from carp (Cyprinus carpio). The article alternated among a review article, a perspective article, and a text about what an ideal structure for GBR would look like. I liked the approach.
Below I will make some suggestions and comments to enrich the work. The evaluation was carried out under the eyes of a researcher familiar with the synthesis and characterization of biomaterials, but whose specialty is in vitro and in vivo assays.
Introduction
The introduction contextualizes the problem well. I have little to contribute to this.
A point that I consider relevant, between lines 29 and 34 the authors show some limitations of the materials currently available for use in humans. There is also no so-called “gold standard” for bone implants, which justifies the search for new materials and methodologies. Maybe it's worth mentioning this in the text.
Among lines 44 and 50 the authors describe their objectives with this manuscript. As indicated, this review aims to validate the material under study. I think it is okay, as long as the favorable and contrary results are presented clearly and objectively throughout the text. I think the advantages are well explored. I suggest pointing out some of the limitations as well.
Literature review
At this point, various TERM parameters applied to bone will be discussed. Overall, the text is good. Below are some suggestions for authors to contribute:
Among lines 81 and 122 the general mechanisms of bone repair are discussed. The approach is simple, but I actually don't think it needs more depth, given the objectives of the article.
In item 2.1.3 (Necessary conditions for successful bone regeneration procedures with a focus on guided regeneration (ROG)), mainly among lines 127 and 133, the authors discuss important terms such as bone healing (repair) and bone regeneration. I don't want to be controversial, but the terms healing, repair and regeneration are very well defined in Pathology books. This underlies almost all-medical sciences. For example, look at the definitions I find in the book by Kumar et al 2020 (Robbins & Cotran Pathologic Basis of Disease, p 103), which the authors certainly know:
- Repair, also called healing, refers to the restoration of tissue architecture and function after an injury.
- Repair of damaged tissues occurs by two processes: regeneration, which restores normal cells, and scarring, the deposition of connective tissue
- Regeneration. Some tissues are able to replace the damaged components and essentially return to a normal state; this process is called regeneration (Kumar et al 2020, p 103).
Using different definitions can be a problem. Therefore, I don't know if the terms used are variations used in orthopedics. However, this needs to be clear. If this is the case, I suggest that the authors say: “in this text we will adopt the appropriate nomenclature to describe healing, repair and biological regeneration” or something similar. Alternatively, the authors could pay attention to the most common nomenclature.
Next, a long text comes about materials for GBR. I liked the text.
Another suggestion is that, in this part of the manuscript, some concepts are brought up, such as “Biocompatibility”, “Bioactivity”, “resorbable”, “functional”, “Biodegradability”. I suggest the authors consult Vert et al. Pure Appl Chem 84 (2): 377-410, 2012 (http://dx.doi.org/10.1351/PAC-REC-10-12-04). This can greatly enrich the text, at the discretion of the authors.
I enjoyed the discussion on pore size variation (Macroporous Structure, lines 369-376). This shows how much we still don't know about the implications of pores for bones. A point that may be worth a little more clarification is that, normally, “pore size” is closely related to tissue adhesion and growth. In the case of bone, in practice, the interpretation of “ideal for new bone formation” is more common. Perhaps the authors can make this distinction. This helps make the numbers even more open. I think this album is worth it.
In the version that came to me, there is a large space at the end of page 11 (end of line 405). I don't know if something was lost. I don't think so, but it's worth checking out.
Between lines 494 and 508 (Chemical cross-linking) the authors discuss cross-linking agents for collagen. Fortunately or unfortunately, glutaraldehyde is still a widely used agent. Both experimentally and for commercial products. I believe that some lines dedicated to him are necessary.
Among lines 522 and 526 there is a discussion about the use of Electospinning to form collagen fibers. The formation of fibers from natural polymers is challenging. In addition to preserving the collagen structure, another sensitive point is achieving large-scale production in a relatively short time. Maybe Electospinning isn't the best method for this.
Discussion
The following are data on a patented collagen material obtained from carp (Cyprinus carpio). Very interesting. However, it seems that this material has no limitations. I believe that the authors can defend their point of view, but they say that the membrane in question still needs more studies to prove its efficiency. Personally, based on the data presented, it looks quite promising.
Conclusion
The conclusions were careful. I agree with them. I understand that this manuscript is organized more as a defense of a point of view than as a review. As I said, I think this is corect. I just suggest reviewing the points mentioned above.
Round 2
Reviewer 2 Report
Comments and Suggestions for Authors
The manuscript structure has improved after revision.
Comments on the Quality of English LanguageNo editing of English language required.